# Technological Improvements on FML in the Chianti Classico Wine Production: Co-Inoculation or Sequential Inoculation?

**DOI:** 10.3390/foods11071011

**Published:** 2022-03-30

**Authors:** Alessandro Bianchi, Isabella Taglieri, Francesca Venturi, Chiara Sanmartin, Giuseppe Ferroni, Monica Macaluso, Fabrizio Palla, Guido Flamini, Angela Zinnai

**Affiliations:** 1Department of Agriculture, Food and Environment, University of Pisa, Via del Borghetto 80, 56124 Pisa, Italy; alessandro.bianchi@phd.unipi.it (A.B.); isabella.taglieri@for.unipi.it (I.T.); francesca.venturi@unipi.it (F.V.); chiara.sanmartin@unipi.it (C.S.); giuseppe.ferroni@unipi.it (G.F.); angela.zinnai@unipi.it (A.Z.); 2Interdepartmental Research Centre “Nutraceuticals and Food for Health”, University of Pisa, Via del Borghetto 80, 56124 Pisa, Italy; 3National Institute of Nuclear Physics (INFN), Sezione di Pisa, Largo Bruno Pontecorvo, 3, 56127 Pisa, Italy; fabrizio.palla@cern.ch; 4Department of Pharmacy, University of Pisa, Via Bonanno Pisano 6, 56126 Pisa, Italy; guido.flamini@unipi.it

**Keywords:** co-inoculation, sequential inoculation, *Oenococcus oeni*, *Lactobacillus plantarum*

## Abstract

Winemaking variables and techniques are known to affect the composition of wines. To obtain a rapid and safe fermentation course, with a reduced risk of proliferation of unwanted microbial species, frequent recourse is made to the use of selected microorganisms, which can greatly simplify the complex management of the fermentation process. In particular, selected strains of lactic acid bacteria are used, which are much more sensitive than yeasts to the operating conditions of the medium. In this regard, the overall aim of this research was to verify whether the early inoculation of homolactic acid bacteria for hexoses (*Lactobacillus plantarum*) carried out after 24 h, compared with that of saccharomycetes operating alcoholic fermentation, could be advantageous compared with a traditional innoculation with a different heterolactic bacterial strain for hexoses (*Oenococcus oeni*) operated at the end of alcoholic fermentation. The grape variety chosen was Sangiovese, the protagonist of Tuscan oenology. The evaluation focused on different aspects such as the management of winery operations, and the quality and longevity of the product; was carried out in all phases of winemaking; and analysed both from a chemical and sensory point of view.

## 1. Introduction

Most red wines, but also some white wines and basic sparkling wines, undergo malolactic fermentation (MLF) to improve wine organoleptic properties and provide further improvements. Traditionally, *Oenococcus oeni* has been chosen to carry out MLF [1,2,3], thanks to its ability to tolerate the unfavourable environment of wine after the end of alcoholic fermentation [4]. However, exploration of relevant oenological characteristics of *Lactobacillus plantarum*, such as tolerance to harsh wine conditions similar to *Oenococcus oeni*, and a more diverse array of enzymes for wine aroma and colour changes [5], contribute to the arrival of *L. plantarum* as the new generation of wine MLF starter cultures [6,7,8].

Several works on changes in the chemical and organoleptic properties of wine are based on the comparison of *O. oeni* and *L. plantarum* for their metabolic potentials [9,10,11,12], the timing of inoculation [10,13,14] or yeast interaction, with each of the two bacterial strains [15,16,17]. All previous studies have focused on the use of a single MLF culture (either *O. oeni* or *L. plantarum*) for an overall reduction in fermentation time or changes in the volatile and sensory properties of wine [18]. Previous studies have shown that different strains with different metabolic activities could alter the wine characteristics, depending on the fermentation parameters [19,20].

Limited studies have focused on a blend of both *L plantarum* and *O. oeni* on wine organoleptic properties and fermentation dynamics [7,9]. The studies observed a distinct profile of volatile compounds mainly for anthocyanin and other phenols at various inoculation timings [13]. The differences were considered to be the result of the metabolism of yeast and bacteria in the different stages of wine fermentation [15].

The overall aim of this research is to verify whether the early inoculation of homolactic acid bacteria for hexoses (*L. plantarum*), carried out after 24 h, compared with that of the saccharomycetes operating alcoholic fermentation, could be advantageous with respect to a traditional innoculation with a different heterolactic bacterial strain for hexoses (*O. oeni*) operated at the end of alcoholic fermentation. In particular, the moment at which the starter cultures of lactic acid bacteria are inoculated is an important factor that influences not only the success of malolactic fermentation, in terms of depletion of metabolisable compounds (e.g., malic acid, hexoses, citric acid), but also the management of subsequent cellar operations. In addition, the methods of inoculation affect the time interval during which the must/wine is exposed to microbial and oxidative alterations. The grape variety chosen was Sangiovese, the protagonist of Tuscan oenology and the main component of the seven Tuscan DOCGs: Brunello di Montalcino, Carmignano, Chianti, Chianti Classico, Morellino di Scansano, Montecucco, and Nobile di Montepulciano. Currently, the “Sangiovese” vine is the most widespread vine in Italy, with about 10% of the total vineyard area [21]. In Tuscany, it is the most widespread vine, with 67.4% of the regional wine-growing area [22].

The evaluation focused on different aspects such as the management of winery operations, the quality and longevity of the product, carried out in all phases of winemaking, analysed both from a chemical and sensory point of view.

## 2. Materials and Methods

### 2.1. Raw Materials

The study was conducted in 2019 on wines produced in the Marchesi Antinori cellar (Via Cassia per Siena, 133, 50026 Firenze, Italy), from Sangiovese grapes (*Vitis vinifera*) from 3 vineyards located in 3 areas (C = Castellina, G = Gaiole, and T = Tavarnelle) in the Chianti Classico DOCG region, as previously reported [23]. Table 1 shows the analyses of the grapes previously evaluated [23].

### 2.2. Alcoholic Fermentation

After the harvest, the grapes arrived in the cellar and were destemmed, crushed and positioned inside 51 hectolitre stainless steel tanks. Before the grapes entered the tank, it was decided to saturate the tanks with CO_2_ produced by another tank in the cellar in full fermentation, in order to guarantee maximum protection from oxidation and avoid the addition of sulfur dioxide. The first Babo was carried out and the SAEN 5000 system activated for automatic regulation of temperature and oxygenation according to the vinification protocol shown in Table 2.

For the fermentation, 20 g/hL of Selectys^®^ Italica CR1 yeast (OENOFRANCE Montebello Vicentino (VI), Italy) was used, due to its good and regular fermentation capacity. CR1 yeast does not need a lot of sustenance in terms of assimilable nitrogen, but it was decided to support its development by adding 30 g/hL of Nutriferm^®^ Vit Flo (fermentation activator based on ammonium phosphate (99.8%) and thiamine (0.2%) (Enartis, San Martino (NO), Italy), rationed in several stages during fermentation. The total dose of the tank was decided on the basis of the difference of the last readily assimilable nitrogen value recorded in the grapes before harvest, and 250 mg/L (considered as the ideal value for optimal fermentation). In addition, Nutrient Vit Nature (Lallemand Inc. Italia, Castel D’Azzano (VR), Italy) was also added (derived nutrient entirely from a yeast autolysate particularly rich in organic nitrogen) at mid-fermentation and Nutrient Vit End™ (Lallemand Inc. Italia, Castel D’Azzano (VR), Italy) (nutrient containing a combination of inactive yeasts and cell walls) towards the end of fermentation, to ensure stable fermentation and avoid possible fermentation blocks. The processing of the cap during the alcoholic fermentation was performed manually with a hydraulic piston. The first fulling was carried out about 24 h after filling the tank. Subsequently, and until the end of the fermentation of the sugars, twice daily punching down was carried out. After the alcoholic fermentation, daily fulling and reassembly were carried out until the hat had begun to show signs of subsidence, after which only twice daily pumping over was carried out until the end of maceration, which generally lasted 5–6 days.

### 2.3. Malolactic Fermentation and Aging

The management of malolactic fermentation was the central point of the experimental work, and two different bacteria were examined following their inoculation protocol. The first protocol (samples CP = Castellina plantarum, TP = Tavarnelle plantarum, and GP = Gaiole plantarum) involved the use of *Lactobacillus plantarum* bacteria through an early co-inoculation of 10 g/hL of ML PRIME (Lallemand Inc., Castel D’Azzano VR, Italy) 24 h after mashing. The second protocol (samples C = Castellina, T = Tavarnelle, and G = Gaiole) instead used the bacterium *Oenococcus oeni* through the sequential inoculation of 10 g/hL of Lalvin VP41 (the commercial strain in the second protocol was also supplied by Lallemand Inc. Italia).

After the maceration, the wines of both protocols were placed in barrique (10 for each thesis). The excess drippings and the crushed grapes were no longer be treated in this research as they were combined with other masses present in the cellar. The barrel cellar was kept at a temperature of 17 °C with a humidity of 75% to favour the development of bacteria for about two months during development and malolactic fermentation. After the malolactic fermentation, the temperature of the barrel was lowered to about 14–15 °C and the humidity increased to about 80% to prevent the evaporation of the product and the development of unwanted microorganisms. At the end of the malolactic fermentation (complete exhaustion of malic acid and residual hexoses) the wines were stabilised with SO_2_ to the amount of about 0.8–0.9 mg/L of molecular SO_2_. The dosage of potassium metabisulphite varied according to the matrix on which it was dosed (pH, alcohol and temperature). The molecular SO_2_ content of all the barriques was maintained between 0.8 and 0.9 mg/L until 31 March 2020. In the first week of April the wines from the barriques of each area were reassembled, filtered and finally repositioned in their own barriques, which, in the meantime, had been adequately cleaned with steam. At the end of June 2020, a reassembly of the masses was carried out again. The aging ended after 17 months (April 2021), when all the barriques for each thesis were reassembled separately and the wine bottled.

### 2.4. Chemical Analyses

All chemical determinations necessary for the characterisation of wines (sugar content (hexoses g/L), titratable acidity (tartaric acid g/L), pH, L-malic acid (g/L) and potassium (mg/L)) were performed as previously described [23]. The other determinations were carried out according to OIV methods [24], in particular, alcohol content (%*v*/*v*) following the OIV-MA-AS312-01A, AVN (net volatile acidity (g/L acetic acid) following the OIV-MA-AS313-02A, dry extract (g/L) following the OIV-MA-AS2-03A, and ash (g/L) following the OIV-MAAS2-04A; however, total polyphenols (g/L catechins), total anthocyanins (g/L malvin) and decolourable anthocyanins (g/L malvin) were measured as previously reported [25].

For determination of the chromatic characteristics, a Benchtop CLM-196 colorimeter (Eoptis-38121, Trento (TN), Italy) was used. The acquired colour values were expressed using the native CIE L* C *h *coordinates as previously reported by [26].

### 2.5. Volatile Compounds (VOCs) SPME-GC/MS

The aromatic profile of wine samples was determined following the protocol previously described [27], using a SPME Supelco (DVB/CAR/PDMS 50/30 μm coating thickness, St. Louis, MO, USA) for the sampling of volatiles, followed by gas chromatography–electron impact mass spectrometry (GC–EIMS) (Agilent Technologies Inc., Santa Clara, CA, USA).

### 2.6. Sensory Analysis

The “expert panel” of the DAFE constituted 10 assessors (4 males and 6 females, 23–60 years), selected and trained as described [28], who performed the sensory analysis, following the procedure described [27].

### 2.7. Statistical Analyses

One-way ANOVA was run (CoStat, Version 6.451, CoHort Software, Pacific Grove, CA, USA) and Tukey’s honestly significant difference (HSD) test, with *p* ≤ 0.05 for multiple comparison, was used.

In order to evaluate the kinetics trend, a C++ program based on a Root framework [29] was used for data processing as previously described [23].

Sensory analysis results were processed by Big Sensory Soft 2.0 (version 2018). In particular, sensory data were analysed by two-way ANOVA with panelists and samples as the main factors [30].

## 3. Results and Discussion

### 3.1. Conversion of Sugars during Alcoholic Fermentation

Fermentation determines the exhaustion of the hexoses in all the musts leading to a reduced carbohydrate residue (≤1 g/L) (Table 3), so the metabolism of the yeasts does not seem to be conditioned by the activity of the homolactic bacteria (*Lactobacillus plantarum*) in early inoculation compared with the heterolactic bacteria (*Oenococcus oeni*) used in sequential inoculation. Figure 1 shows the kinetics of sugars consumption and ethanol production. In accordance with the provisions of the alcoholic fermentation balance, a quantity of ethanol similar to the theoretical process yield (91%) is recorded, highlighting that no significant metabolic deviations occurred both in the case of musts submitted to early inoculation and in those with sequential inoculation [31]. The fermentation trend had a regular course in all theses, with a duration of between 10 and 13 days based on the starting concentration of sugars. 

### 3.2. The Trend of pH, Acid Component and Ash during Alcoholic Fermentation

During the vinification of the six musts, the changes in pH value, titratable acidity, ashes and volatile acidity residue were evaluated to verify the evolution of the acidic profile and the degree of neutralisation of the wines. In accordance with what was already observed for the conversion of hexoses, there was no significant effect related to the activity of bacteria operating during alcoholic fermentation compared with those that were active after the end of yeast activity (Table 4).

In the case of musts characterised by an early inoculation of lactobacilli, a greater difference was observed as a consequence of being linked to the metabolic activity of those microorganisms (metabolisation of malic acid, hexoses residues, etc.) [17,18].

Comparing the data relating to the trends of titratable acidity in musts–wines obtained from the same grapes but with different inoculation methods, a modest decrease was observed, regardless of the activity carried out, or not, by lactic acid bacteria, with a slight difference only in the case of CP wine where the activity of *L. plantarum* determined the achievement of the lowest acidic value among the analysed wines. From the comparison of the wines, it was observed that the acetic acid content was very low and similar with regard to the TP wine which had the lowest value.

From examination of the ashes trend, no significant differences were observed in any of the compared samples. This consideration allowed us to affirm that there were no differences related to problems of tartaric instability resulting from the activity of lactic acid bacteria during alcoholic fermentation [4,32]. 

### 3.3. Malolactic Fermentation during Alcoholic Fermentation (Lactobacillus plantarum)

Malolactic fermentation took place in musts inoculated 24 h after mashing with *L. plantarum*, with complete decarboxylation of malic acid into lactic acid, which favours the possibility of early stabilisation of wines. In the other three musts, the variation of malic acid was very modest, reflecting the role of this compound that represents an intermediate process in the metabolic activity of yeasts [33]. Parallel to the conversion of malic acid, in the wines inoculated with *L. plantarum*, was the synthesis of lactic acid (in its two stereoisomers L and D) (Table 5). The accumulation of lactic acid has the opposite sign to the decrease in malic but a very similar value, as can be seen from the comparison of the data reported in Figure 2. Gaiole had a higher net dry residue as a result of a reduced variation of the destroyable acidic component (e.g., malic acid, citric).

### 3.4. Sequential Malolactic Fermentation (Oenococcus oeni)

In Table 6, it is possible to note how the duration of the malolactic fermentation strongly depended on the type of inoculum [34]. In the three wines with plantarum inoculum, the malolactic fermentation took place almost simultaneously with the alcoholic fermentation, since the inoculation was performed at the beginning of the alcoholic fermentation; conversely, the other three wines were inoculated on racking, extending the fermentation times widely [2]. 

Since malolactic bacteria are able to convert not only malic acid but also other substrates, with different conversion rates, the concentration in lactic acid of wines at the end of aging is also different because the process yield changes according to the type of microorganisms (homolactic or heterolactic) [35,36]. In fact, *L. plantarum* converts hexoses only into lactic acid (100% by weight), while *O. oeni* synthesises other reaction products (e.g., acetic acid, ethanol) in addition to lactic acid (equal to 50% by weight of converted carbohydrates), as shown in Table 6 (Figure 3). 

A comparative analysis of data relating to the type and duration of the metabolic activity of lactic acid bacteria showed the difference in the behaviour of lactic acid bacteria in wines inoculated with *O.*
*oeni*, which converted malic acid with a certain slowness (t > 39 days), particularly in the case of Gaiole (t ≅ 60 days). This difference, partly explained by the rather low pH value of this wine (pH = 3.22) [37,38], assumes a particularly important technical relevance, because of a delay in the sulphation of the product, given the high sensitivity of malolactic bacteria towards this additive [39]. A further consideration useful to understand what was observed, is that only a little more than one third of the lactic product was derived from the conversion of the malic; the remaining part came from sugars that were metabolised much more slowly. Overall, it can be noted that the lactic acid content was higher in the products in which lactobacilli were inoculated, a positive factor for the longevity and stability of wines, as the fixed acidity is a protective element against unwanted microorganisms and tartaric precipitation [32].

### 3.5. The Changes in pH Value, Acid Component and Ash during Aging

In the period following the alcoholic fermentation, the trend of aging was evaluated. From the analysis of the data (Table 7), it was clear that the pH did not change significantly between the tests conducted working with *L.*
*plantarum* and *O. oeni* in the time interval between racking and the entire duration of aging in wood (t = 500 days).

Comparing the data of the products inoculated with *L.*
*plantarum* to those with *O.*
*oeni*, a high acidic content was observed at the end of aging, probably linked to greater chemical-physical stability and a lower incidence of tartaric precipitation [32,40]. However, the final titratable acidity values were all higher than 5 g/L (Table 7), a value consistent with the type of product desired (medium-aged red wine with good structure). 

The fixed acidity values of wines inoculated with *L.*
*plantarum* had higher values (on average +0.4 g/L), more favourable to the compositional characteristics of red wine. Even the ashes tended to reduce over time, with a speed that was not too different. The net volatile acidity showed contained values even at the end of aging, especially in wines inoculated with *L.*
*plantarum*, experimental evidence that may be related to the type of metabolism of the hexoses. An analysis of the fixed acidity trend (Figure 4) showed that it tended to decrease over time parallel with the stabilisation of the chemical-physical balances of the wine, according to a linear trend; additionally, in this case, by comparing the salt content of the wines inoculated with *L.*
*plantarum* with the others, higher values were observed, while there were no differences related to the production area of the grapes. It was observed that the dry residue at the end of aging was higher in wines inoculated with *L. plantarum*, which by carrying out a homolactic conversion of sugars did not determine mass loss from this type of substrate even if the differences were modest, given the reduced concentration in hexoses at the end of fermentation [36]. This evidence was confirmed, despite the fact that, at the end of racking, the dry residue was significantly higher (on average +1.8 g/L) in the products obtained by conducting the more traditional malolactic fermentation (i.e., at the end of alcoholic fermentation and using a heterolactic fermenter, such as *O. oeni*).

### 3.6. The Trend in Phenol Concentration during Aging 

With regard to phenols content (Table 8), it was observed that the phenolic concentration at the end of fermentation was similar in wines in which early inoculation was adopted, compared with the others. However, a modest but significant difference was observed for samples from the T vineyard, in which the content of total polyphenols was higher than that in which lactobacilli were inoculated. 

The content of total phenols in wines inoculated with *L. plantarum* seemed more favourable in two of the three products analysed. In fact, while for wines C and T, the total phenolic content was significantly higher than that of the corresponding samples vinified in a more traditional way; in the case of G, the wines were characterised by the same content in total phenols [5,41]. The different behaviour was ascribed to the different stability of the individual phenolic groups [42]. Indeed, anthocyanins undergo a decrease during the period of aging linked to the degradative reactions in which they are involved (oxidation and solubilisation). In addition, with regard to the evolution of bleachable anthocyanins in wines from the different areas covered by the experimentation, we saw a trend similar to that observed for the total phenols, confirming the different chromatic instability was linked to the characteristics of the raw material used in winemaking.

In general, it was possible to observe (Table 8) a significant reduction in the unstable chromatic component (−33%) in all wines, regardless of the lactic acid bacteria used or the area of origin of the grapes. However, it is possible to note that this decrease was significantly higher in wines in which lactobacilli had been inoculated prematurely (−13%), therefore, the colour should be more stable and more intense in these products as was evidenced by the sensory evaluations (Section 3.7). 

The chromatic parameters of wines at the end of aging are shown in Table 9. Comparing the data of the wines obtained from the same grapes using different lactic ferments, differences were found due to the activity of these microorganisms [41], because there was a different decrease in the colour product, despite the fact that at the end of the aging phase they had very similar concentrations of total anthocyanins. This observation seems to indicate a different stability of the chromatic component present in the different wines. 

### 3.7. Volatile Compounds VOCs

The volatile fraction was analysed by head-space SMPE during the different phases of winemaking, allowing the identification of some classes of compounds (acids, alcohols, esters) present both at racking (Table 10) and at the end of aging (Table 11). The most significant compounds of the individual classes present were phenyl-ethyl alcohol, *iso*-pentyl alcohol and 2 methyl-butanol, which are characterised by spicy and toasted notes and are strongly influenced both by the variety of grapes and the type of aging [23]. The main esters were ethyl octanoate, ethyl decanoate, ethyl hexanoate, isoamyl acetate, ethyl decanoate, which are characterised by fruity-floral scents.

From racking to the end of aging these two classes of compounds tended to have an opposite trend, in fact, esters tended to decrease over time, while alcohols/phenols and aldehydes/ketones tended to increase.

From the statistical analysis (PCA) (Figure 5) it is possible to note that at the time of racking, the wines were associated in two different groups according to the area of origin; in fact, the wines from the areas of C and G were similar from the aromatic point of view regardless of the protocol adopted, while they differed considerably from the wines from the T area. However, PCA was able to separate C and G wines, too. If we go into more detail regarding the two different groups, it was possible to note that at the end of the alcoholic fermentation there were no differences within the same area between the wines produced with co-inoculation or sequential inoculation, demonstrating that the effect of the area, and on the technology adopted, in these phases, is prevalent. 

From the hierarchical cluster analysis (Figure 5), it is possible to notice how at the time of racking (t = 1) the wines were associated in two different groups; through the analysis of the PCA (Figure 6) it is possible to note that there were no differences within the same area, at the end of the alcoholic fermentation, between the wines produced with co-inoculation and sequential inoculation. The hierarchical analysis (Figure 5) carried out at the end of aging (t = 6) showed that wines tended to associate in three groups, independent of the area of origin of the grapes, denoting that the wine-making technology had greatly influenced the odorous expression of the wines. The product that tended to differentiate itself most from the other wines because it had a clearly different aromatic profile, was the Tavarnelle plantarum, as was confirmed by the sensory evaluation. From the comparison of the PCAs carried out at the racking and at the end of aging, it is possible to highlight that the wines produced with grapes of the same area were initially placed in the same quadrant, which was different for the different areas, regardless of the mode of conduction of malolactic fermentation. On the contrary, at the end of the aging in wood, the different wines were placed differently in the graph obtained, regardless of both the production area of the grapes and the different inoculation of lactic acid bacteria, indicating that the wines had very different compositional characteristics of the odorous component.

### 3.8. Sensory Analysis of Wines

In accordance with what was recorded by the chemical characterisation, among all the parameters considered, the use of different strains of lactic acid bacteria for the conduction of FML seemed to have had the greatest influence on the visual/aromatic component and acidity. At the panel test, wines produced using *L.*
*plantarum* generally had a significantly higher colour intensity and fixed acidity (Figure 7) that remained evident throughout the entire period considered [41].

Figure 8 shows the data relating to the hedonic parameters evaluated by the judges. In general, a trend could be outlined in favour of wines produced with *L.*
*plantarum* in terms of visual pleasantness, while at the end of aging it was possible to highlight significant differences both in terms of olfactory pleasantness and overall pleasantness, depending on the strain used.

## 4. Conclusions

According to the results obtained, it was possible to affirm that the presence of lactobacilli during alcoholic fermentation did not negatively interfere with the extraction of the colourless phenolic component, which had a higher concentration in two of the three wines analysed. This consideration maintained its validity even at the end of aging in wood, because, even at this stage, the phenolic content subtracted from the anthocyanins was higher or equal in wines obtained by early inoculation of malolactic bacteria. However, it was concluded that the effect related to the different conduction of malolactic fermentation, while affecting the structural characteristics of the products, linked to the phenolic components, did not modify them. In accordance with what was recorded by the chemical characterisation, in addition to the sensorial profile, the use of different strains of lactic acid bacteria for the conduction of FML seems to have had the greatest influence on the visual/aromatic component and acidity.

## Figures and Tables

**Figure 1 foods-11-01011-f001:**
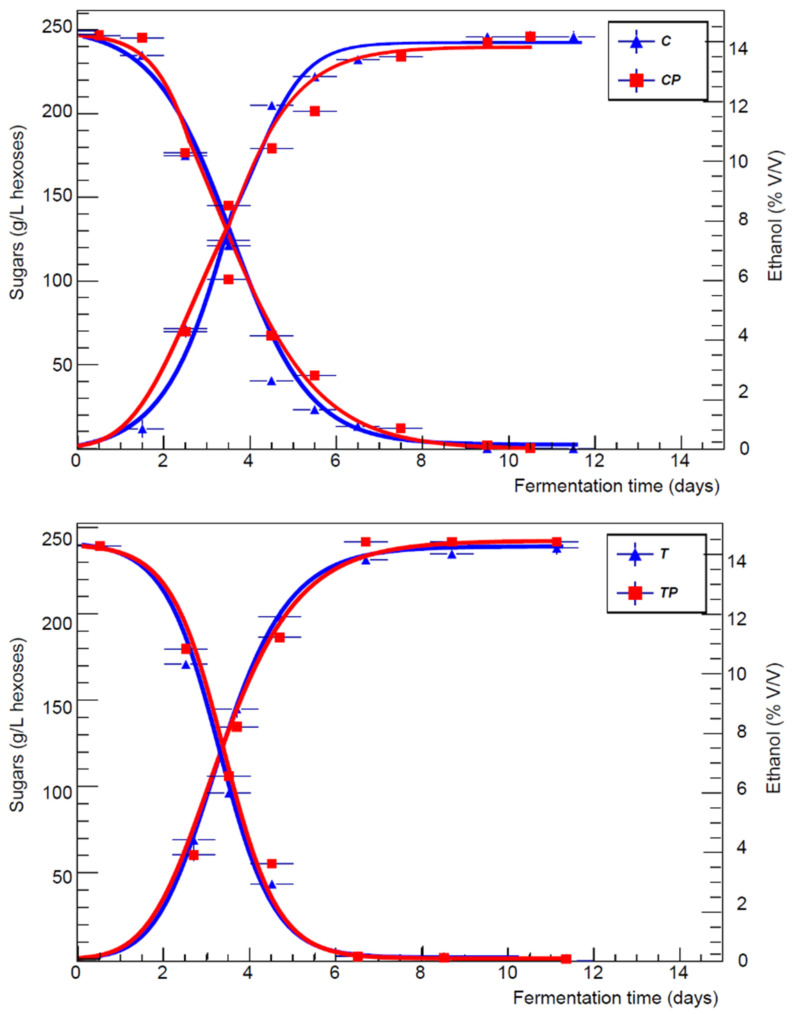
The kinetics of sugars consumption and ethanol production as a function of the fermentation time (days) for the three vineyards.

**Figure 2 foods-11-01011-f002:**
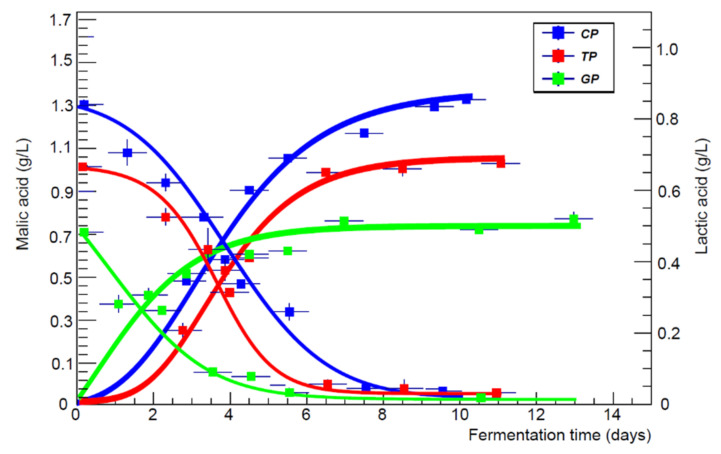
The trend of malic and lactic acid as a function of the fermentation time (days) for all the three plantarum wines.

**Figure 3 foods-11-01011-f003:**
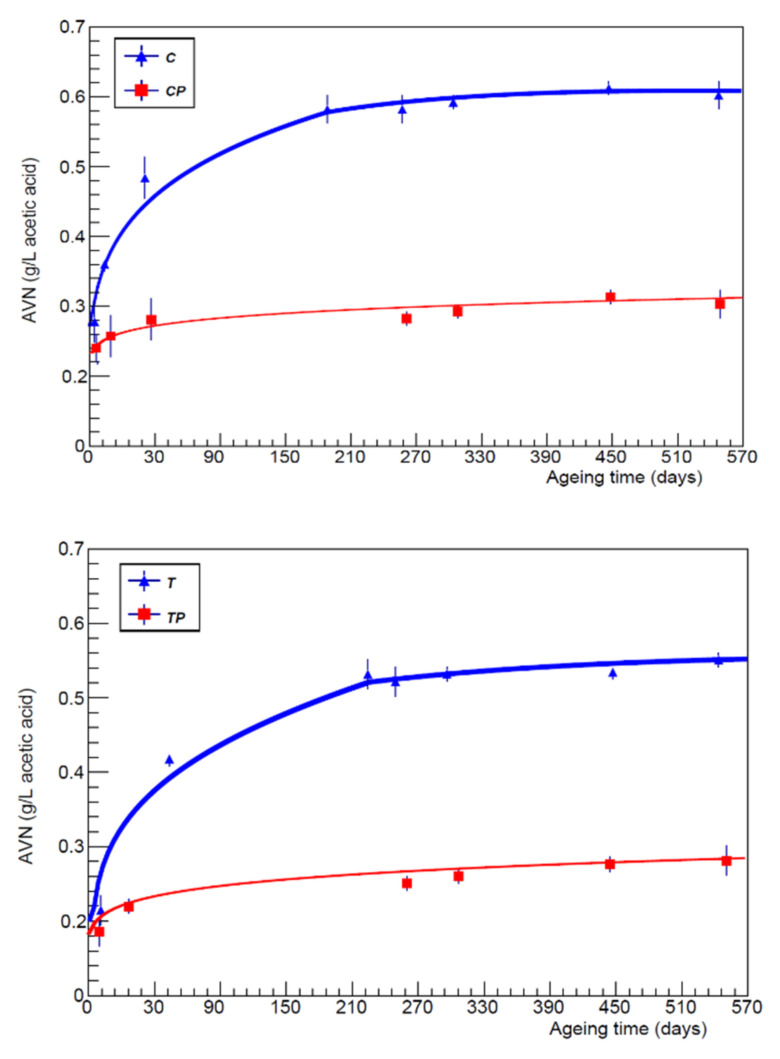
The AVN trend as a function of the aging time (days) for each vineyard and in both protocols.

**Figure 4 foods-11-01011-f004:**
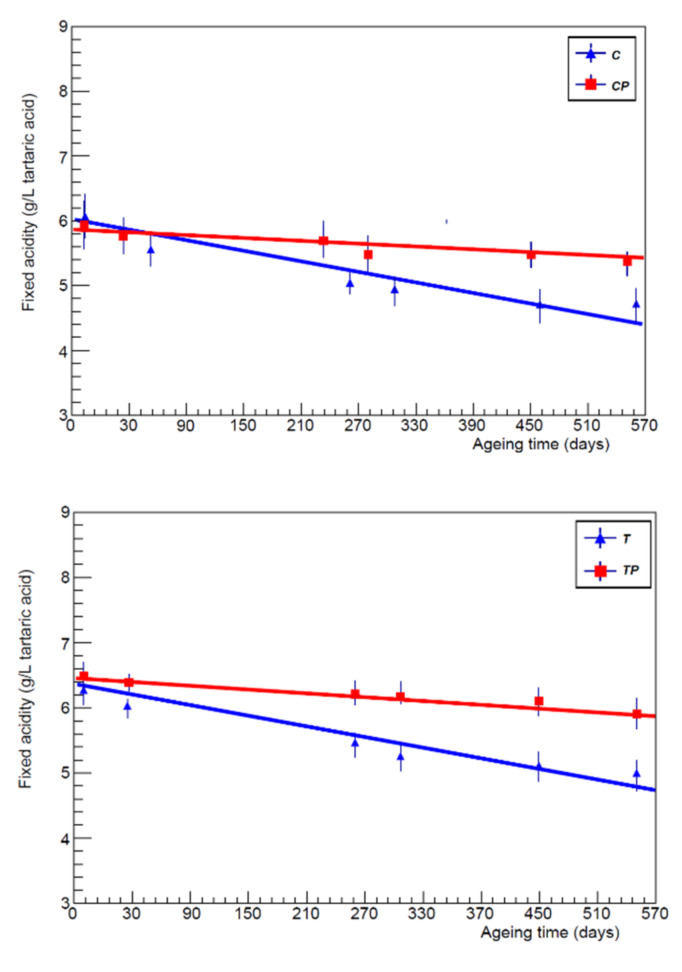
Fixed acidity trend as a function of the aging time (days) for each vineyard and in both protocols.

**Figure 5 foods-11-01011-f005:**
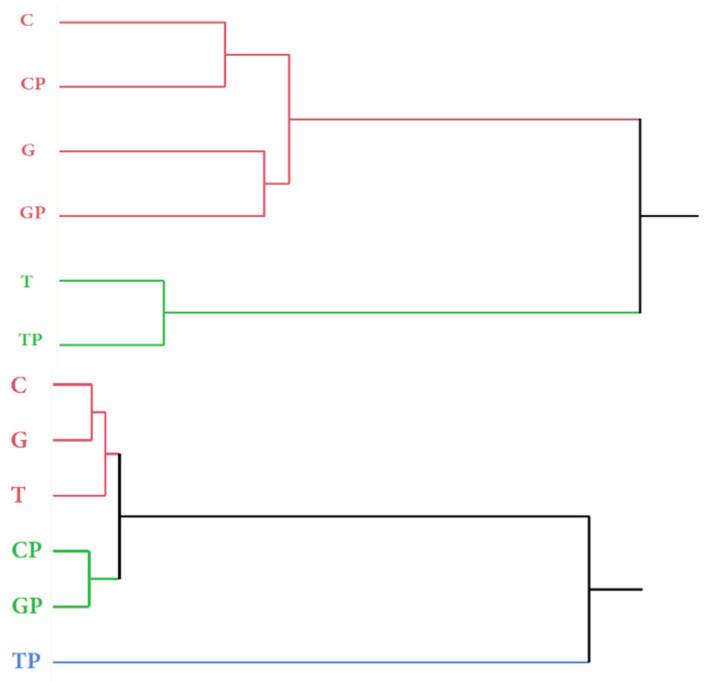
HCA of the GC-MS data relating to wines from the two different protocols analysed at racking at ends of aging.

**Figure 6 foods-11-01011-f006:**
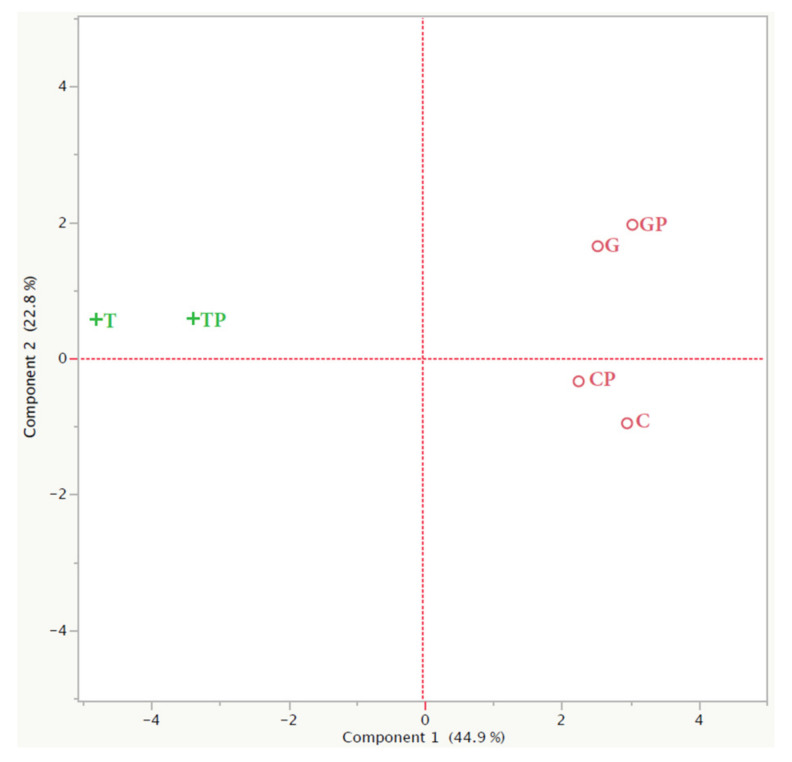
PCA of the GC-MS data relating to wines from the 3 vineyards and vinified according to the two different protocols analysed at racking and at ends of aging.

**Figure 7 foods-11-01011-f007:**
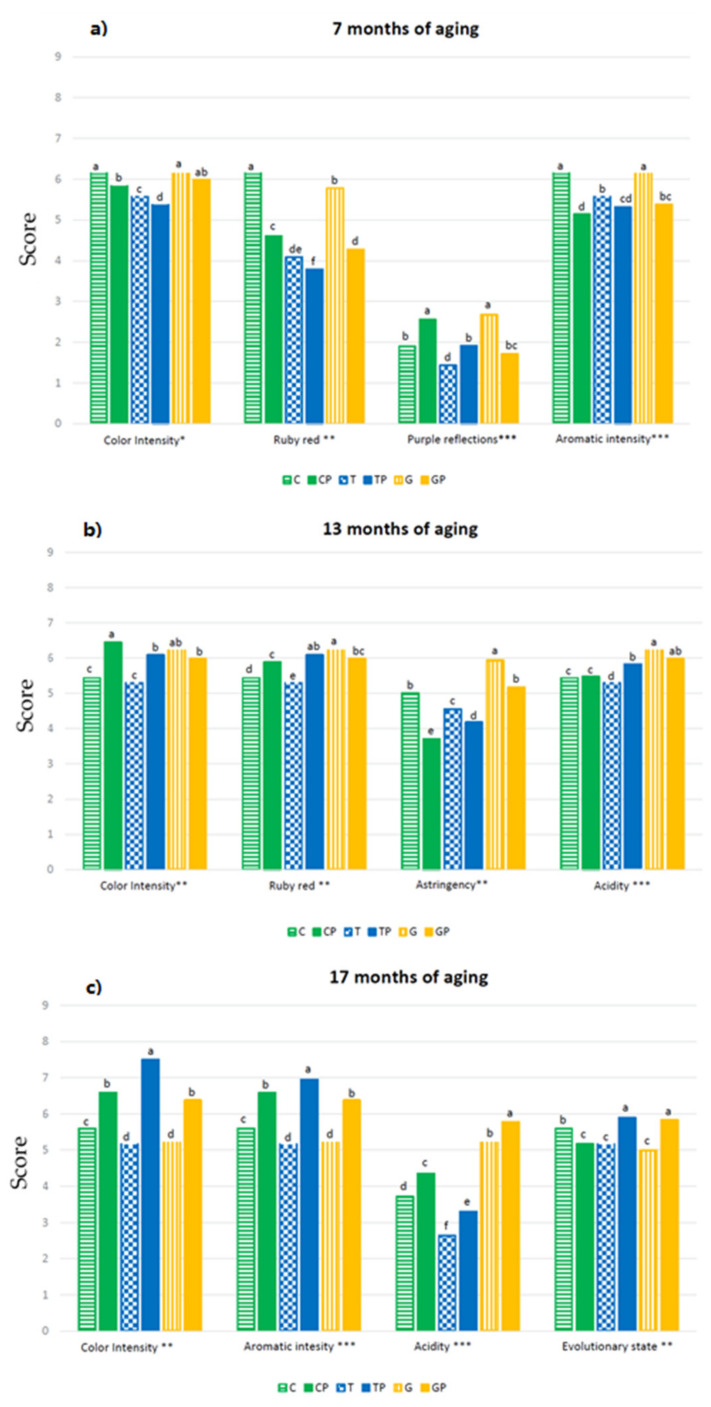
Significant quantitative parameters of wines after: (**a**) 7 months of aging (May 2020), (**b**) 13 months of aging (November 2020), and (**c**) 17 months of aging (March 2021). The evaluation was carried out using a score (0–10). Different letters indicate statistically significant differences at *p* ≤ 0.05 according to the results of two-way ANOVA. Significance level *** *p* < 0.001, ** *p* < 0.01; * *p* ≤ 0.05.

**Figure 8 foods-11-01011-f008:**
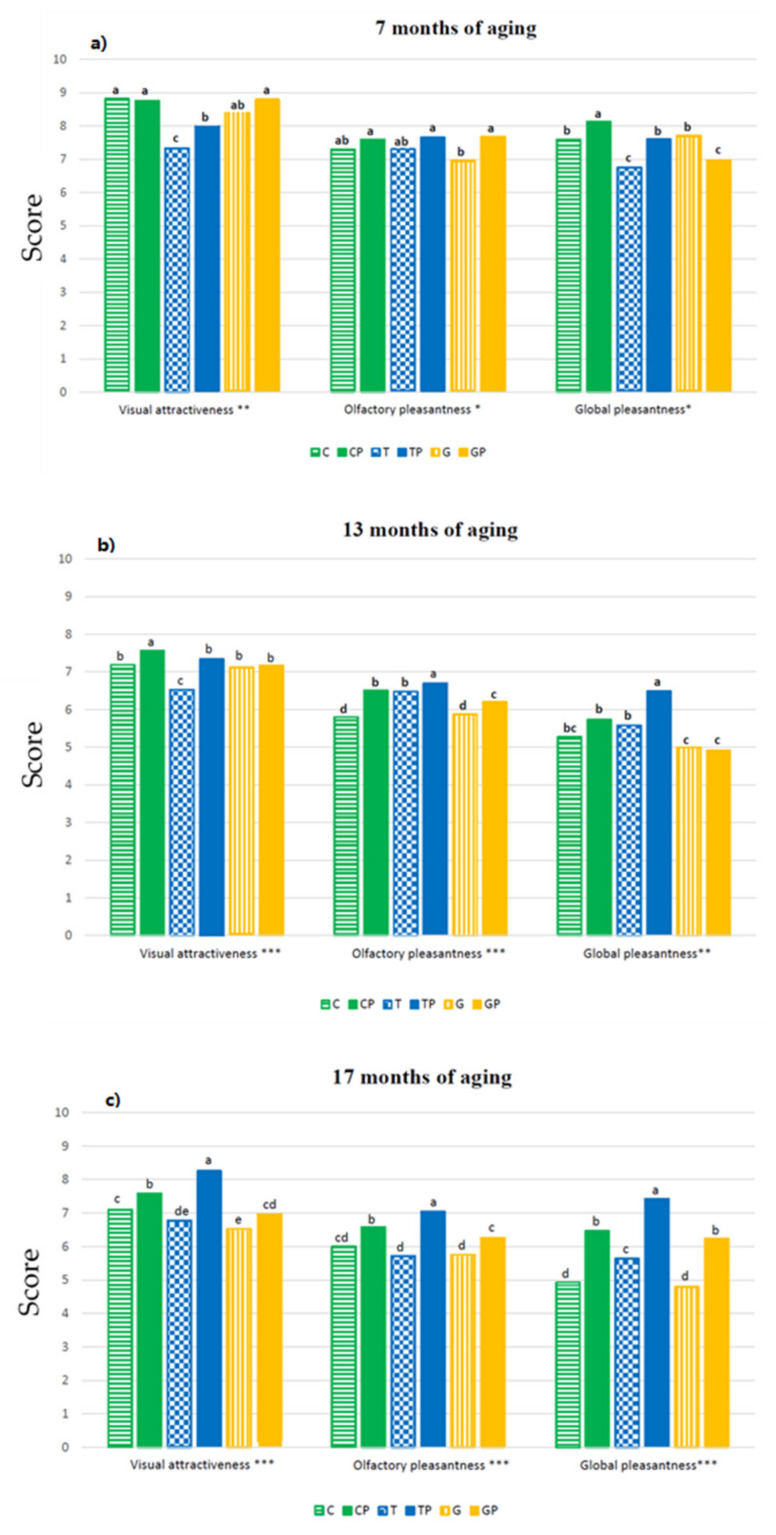
Hedonic parameters of wines after: (**a**) 7 months of aging (May 2020), (**b**) 13 months of aging (November 2020), and (**c**) 17 months of aging (March 2021). The evaluation was carried out using a score (0–10). Different letters indicate statistically significant differences at *p* ≤ 0.05 according to the results of two-way ANOVA. Significance level *** *p* < 0.001, ** *p* < 0.01; * *p* ≤ 0.05; ns: not significant (*p* > 0.05).

**Table 1 foods-11-01011-t001:** Chemical parameters of the grapes at harvest subject of the experimentation. C = Castellina, G = Gaiole, and T = Tavarnelle.

Grape	Sugar Content (Hexoses g/L)	pH	Titratable Acidity (Tartaric Acid g/L)	L-Malic Acid (g/L)	Potassium (mg/L)
C	245 ± 11 a	3.52 ± 0.02 a	6.60 ± 0.10 c	1.29 ± 0.02 a	1284 ± 8 a
T	240 ± 10 a	3.23 ± 0.02 b	7.35 ± 0.12 b	0.99 ± 0.04 b	1188 ± 13 b
G	247 ± 13 a	3.12 ± 0.01 c	8.10 ± 0.13 a	0.70 ± 0.03 c	1005 ± 14 c

Different letters indicate statistically significant differences at *p* ≤ 0.05 according to the results of one-way ANOVA and Tukey’s HSD test. Values are the mean of three technical replicates ± SD.

**Table 2 foods-11-01011-t002:** Oxygenation and temperature protocol used.

Temperature	Oxygen
For 24 h at 26 °C	From the beginning of fermentation up to a drop of 6 °Babo: inactive oxygenation
From 24 h to about 4th Babo: 24 °C	Up to 6 °Babo: 8 mg/L * day
From 4th Babo until the end of fermentation: 28 °C	Up to 1st Babo: 6 mg/L * day
Until racking: room temperature	Until racking: 3 mg/L * day
Barrel cellar: 14 °C and 80% relative humidity	

**Table 3 foods-11-01011-t003:** Values of starting sugars, residual sugars, the duration of fermentation (days), the final ethanol produced, and the % of hexoses converted into ethanol. C = Castellina, G = Gaiole, T = Tavarnelle, CP = Castellina plantarum, GP = Gaiole plantarum, and TP = Tavarnelle plantarum.

Sample	Starting Sugars (g/L Hexoses)	Residual Sugars (g/L Hexoses)	Fermentation Duration (days)	Final Ethanol (%*v*/*v*)	Hexoses Converted to Ethanol (%)
C	244.73 ± 1.12 b	0.91 ± 0.03 b	11	14.4 ± 0.2 a	91
CP	244.73 ± 1.12 b	0.83 ± 0.02 c	10	14.4 ± 0.2 a	91
T	239.25 ± 1.08 c	0.86 ± 0.05 c	11	13.8 ± 0.2 a	90
TP	239.25 ± 1.09 c	0.76 ± 0.03 d	11	14.0 ± 0.1 a	91
G	247.20 ± 1.15 a	1.00 ± 0.04 a	13	14.2 ± 0.2 a	89
GP	249.30 ± 1.17 a	0.82 ± 0.02 c	13	14.2 ± 0.1 a	89

Different letters indicate statistically significant differences at *p* ≤ 0.05 according to the results of one-way ANOVA and Tukey’s HSD test. Values are the mean of three technical replicates ± SD.

**Table 4 foods-11-01011-t004:** Values of starting pH, pH at racking, starting titratable acidity, titratable acidity at racking, starting ashes, ashes at racking, and AVN at racking. C = Castellina, G = Gaiole, T = Tavarnelle, CP = Castellina plantarum, GP = Gaiole plantarum, and TP = Tavarnelle plantarum.

Sample	Starting pH	pH at Racking	Starting Titratable Acidity (g/L Tartaric Acid)	Titratable Acidity at Racking (g/L Tartaric Acid)	Starting Ashes (g/L)	Ashes at Racking (g/L)	AVN at Racking (g/L Acetic Acid)
**C**	3.53 ± 0.02 a	3.50 ± 0.01 a	6.60 ± 0.10 c	6.41 ± 0.12 b	3.21 ± 0.11 b	2.83 ± 0.13 a	0.27 ± 0.01 a
**CP**	3.52 ± 0.02 a	3.50 ± 0.04 a	6.61 ± 0.12 c	6.33 ± 0.17 b	3.22 ± 0.12 b	2.72 ± 0.12 a	0.24 ± 0.02 ab
**T**	3.23 ± 0.01 b	3.29 ± 0.02 b	7.30 ± 0.12 b	6.52 ± 0.15 b	3.24 ± 0.14 b	2.54 ± 0.17 a	0.21 ± 0.02 b
**TP**	3.22 ± 0.01 b	3.30 ± 0.02 b	7.32 ± 0.12 b	6.74 ± 0.14 b	3.28 ± 0.08 b	2.73 ± 0.13 a	0.18 ± 0.02 b
**G**	3.12 ± 0.03 c	3.22 ± 0.03 c	8.09 ± 0.11 a	7.53 ± 0.13 a	3.65 ± 0.15 a	2.82 ± 0.11 a	0.28 ± 0.02 a
**GP**	3.11 ± 0.03 c	3.22 ± 0.04 c	8.11 ± 0.11 a	7.56 ± 0.12 a	3.61 ± 0.11 a	2.96 ± 0.06 a	0.26 ± 0.02 a

Different letters indicate statistically significant differences at *p* ≤ 0.05 according to the results of one-way ANOVA and Tukey’s HSD test. Values are the mean of three technical replicates ± SD.

**Table 5 foods-11-01011-t005:** Values of starting malic acid, malic acid at racking, lactic acid at racking, starting net dry residue, and net dry residue at racking. C = Castellina, G = Gaiole, T = Tavarnelle, CP = Castellina plantarum, GP = Gaiole plantarum, and TP = Tavarnelle plantarum.

Sample	Starting Malic Acid (g/L)	Malic Acid at Racking (g/L)	Lactic Acid at Racking (g/L)	Starting Net Dry Residue (g/L)	Net Dry Residue at Racking (g/L)
**C**	1.28 ± 0.03 a	1.24 ± 0.04 a	0.01 ± 0.01 d	30.3 ± 0.6 a	18.2 ± 0.3 c
**CP**	1.29 ± 0.03 a	0.07 ± 0.02 d	0.87 ± 0.01 a	30.3 ± 0.6 a	17.7 ± 0.3 c
**T**	0.98 ± 0.04 b	0.96 ± 0.05 b	0.02 ± 0.01 d	30.4 ± 0.3 a	18.0 ± 0.4 c
**TP**	1.00 ± 0.03 b	0.08 ± 0.03 d	0.64 ± 0.01 b	30.4 ± 0.3 a	17.1 ± 0.3 c
**G**	0.70 ± 0.05 c	0.66 ± 0.06 c	0.02 ± 0.01 d	29.0 ± 0.4 b	22.7 ± 0.5 a
**GP**	0.71 ± 0.02 c	0.03 ± 0.02 d	0.48 ± 0.01 c	29.0 ± 0.2 b	20.0 ± 0.2 b

Different letters indicate statistically significant differences at *p* ≤ 0.05 according to the results of one-way ANOVA and Tukey’s HSD test. Values are the mean of three technical replicates ± SD.

**Table 6 foods-11-01011-t006:** Lactic acid at the end of malolactic fermentation, lactic acid from malic, lactic acid from heterolactic, lactic acid from homolattic, Δ hexoses (difference between initial and final hexoses), Δ AVN, and Malolactic fermentation duration (days). C = Castellina, G = Gaiole, T = Tavarnelle, CP = Castellina plantarum, GP = Gaiole plantarum, and TP = Tavarnelle plantarum.

Sample	Lactic Acid (g/L) at the End of Malolactic Fermentation	Lactic Acid from Malic (%)	Lactic Acid from Heterolactic (%)	Lactic Acid from Homolattic (%)	Δ Hexoses (g/L)	Δ Avn (g/L Acetic Acid)	Malolactic Fermentation Duration (Days)
C	1.28 ± 0.01 c	64.8	35.2	-	0.91 ± 0.03 b	0.31 ± 0.01 a	39
CP	1.69 ± 0.01 a	51.5	-	48.5	0.83 ± 0.02 c	0.03 ± 0.02 b	15
T	1.07 ± 0.02 d	59.8	40.2	-	0.86 ± 0.05 c	0.29 ± 0.01 a	47
TP	1.39 ± 0.03 b	46.0	-	54	0.76 ± 0.03 d	0.02 ± 0.01 b	17
G	0.94 ± 0.01 e	46.8	53.2	-	1.00 ± 0.04 a	0.34 ± 0.02 a	60
GP	1.31 ± 0.04 bc	36.6	-	63.4	0.82 ± 0.02 c	0.03 ± 0.02 b	19

Different letters indicate statistically significant differences at *p* ≤ 0.05 according to the results of one-way ANOVA and Tukey’s HSD test. Values are the mean of three technical replicates ± SD.

**Table 7 foods-11-01011-t007:** Values of pH at racking, final pH, titratable acidity at racking, final titratable acidity, ashes at racking, final ashes, AVN at racking, fixed acidity at racking, final fixed acidity, net dry residue at racking, and final net dry residue. C = Castellina, G = Gaiole, T = Tavarnelle, CP = Castellina plantarum, GP = Gaiole plantarum, TP = Tavarnelle plantarum.

Sample	pH at Racking	Final pH	Titratable Acidity at Racking (g/L Tartaric Acid)	Final Titratable Acidity (g/L Tartaric Acid)	Ashes at Racking (g/L)	Final Ashes (g/L)	AVN at Racking (g/L Acetic Acid)	Final AVN (g/L Acetic Acid)	Fixed Acidity at Racking (g/L Tartaric Acid)	Final Fixed Acidity (g/L Tartaric Acid)	Net Dry Residue at Racking (g/L)	Final Net Dry Residue (g/L)
C	3.50 ± 0.01 a	3.48 ± 0.03 a	6.41 ± 0.12 b	5.52 ± 0.12 d	2.83 ± 0.13 a	2.01 ± 0.11 ab	0.27 ± 0.01 a	0.59 ± 0.02 ab	6.07 ± 0.11 c	4.77 ± 0.14 d	18.2 ± 0.3 c	16.1 ± 0.1 cd
CP	3.50 ± 0.04 a	3.49 ± 0.03 a	6.33 ± 0.17 b	5.73 ± 0.14 cd	2.72 ± 0.12 a	2.12 ± 0.12 a	0.24 ± 0.02 ab	0.29 ± 0.02 cd	5.99 ± 0.14 c	5.35 ± 0.13 c	17.7 ± 0.3 c	16.9 ± 0.2 c
T	3.29 ± 0.02 b	3.35 ± 0.03 b	6.52 ± 0.15 b	5.85 ± 0.11 c	2.54 ± 0.17 a	1.93 ± 0.13 b	0.21 ± 0.02 b	0.52 ± 0.03 b	6.24 ± 0.14 c	5.16 ± 0.11 c	18.0 ± 0.4 c	15.9 ± 0.2 d
TP	3.30 ± 0.02 b	3.31 ± 0.03 b	6.74 ± 0.14 b	6.23 ± 0.10 b	2.73 ± 0.13 a	2.24 ± 0.14 a	0.18 ± 0.02 b	0.24 ± 0.02 d	6.51 ± 0.15 b	5.89 ± 0.13 b	17.1 ± 0.3 c	16.5 ± 0.3 c
G	3.22 ± 0.03 c	3.29 ± 0.03 b	7.53 ± 0.13 a	6.44 ± 0.12 b	2.82 ± 0.11 a	1.92 ± 0.12 b	0.28 ± 0.02 a	0.63 ± 0.02 a	7.15 ± 0.10 a	5.62 ± 0.14 b	22.7 ± 0.5 a	18.3 ± 0.2 b
GP	3.22 ± 0.04 c	3.27 ± 0.03 b	7.56 ± 0.12 a	6.82 ± 0.13 a	2.96 ± 0.06 a	2.23 ± 0.13 a	0.26 ± 0.02 a	0.32 ± 0.02 c	7.18 ± 0.11 a	6.41 ± 0.12 a	20.0 ± 0.2 b	18.9 ± 0.1 a

Different letters indicate statistically significant differences at *p* ≤ 0.05 according to the results of one-way ANOVA and Tukey’s HSD test. Values are the mean of three technical replicates ± SD.

**Table 8 foods-11-01011-t008:** Chemical parameters at racking. C = Castellina, G = Gaiole, T = Tavarnelle, CP = Castellina plantarum, GP = Gaiole plantarum, and TP = Tavarnelle plantarum.

Sample	Total Polyphenols at Racking (g/L Catechins)	Final Total Polyphenols (g/L Catechins)	Total Anthocyanins at Racking (g/L Malvin)	Final Total Anthocyanins (g/L Malvin)	Decolourable Anthocyanins at Racking (g/L Malvin)	Final Decolourable Anthocyanins (g/L Malvin)	Anthocyanins Ratio at Racking (%)	Final Anthocyanins Ratio (%)
C	4.42 ± 0.03 a	3.32 ± 0.03 c	0.55 ± 0.02 b	0.25 ± 0.01 b	0.42 ± 0.02 b	0.14 ± 0.02 ab	76.4 ± 0.1 b	56.0 ± 0.2 a
CP	4.29 ± 0.04 b	4.06 ± 0.03 a	0.48 ± 0.01 c	0.25 ± 0.01 b	0.34 ± 0.01 c	0.11 ± 0.01 b	70.8 ± 0.1 e	44.0 ± 0.1 e
T	3.75 ± 0.04 cd	2.98 ± 0.02 e	0.38 ± 0.02 d	0.22 ± 0.01 c	0.30 ± 0.01 d	0.11 ± 0.01 b	78.9 ± 0.1 a	50.1 ± 0.1 c
TP	3.55 ± 0.04 e	3.13 ± 0.04 d	0.34 ± 0.01 e	0.21 ± 0.02 c	0.25 ± 0.01 e	0.10 ± 0.01 b	73.5 ± 0.1 c	45.5 ± 0.2 d
G	3.85 ± 0.03 c	3.33 ± 0.03 c	0.66 ± 0.02 a	0.33 ± 0.01 a	0.48 ± 0.02 a	0.18 ± 0.01 a	72.7 ± 0.1 d	54.5 ± 0.2 b
GP	3.65 ± 0.03 d	3.42 ± 0.02 b	0.58 ± 0.03 b	0.31 ± 0.02 a	0.41 ± 0.01 b	0.14 ± 0.02 a	70.6 ± 0.1 e	45.2 ± 0.1 d

Different letters indicate statistically significant differences at *p* ≤ 0.05 according to the results of one-way ANOVA and Tukey’s HSD test. Values are the mean of three technical replicates ± SD.

**Table 9 foods-11-01011-t009:** Wine CIELAB parameters (L*, C*, h*) at the end of aging.

Sample	L*	C*	h*
C	3.05 ± 0.03 b	33.37 ± 0.13 a	12.91 ± 0.12 c
CP	−0.73 ± 0.04 e	28.89 ± 0.13 f	2.52 ± 0.11 e
T	4.02 ± 0.04 a	30.25 ± 0.12 c	19.62 ± 0.11 a
TP	−0.83 ± 0.04 f	27.01 ± 0.14 e	8.14 ± 0.13 d
G	1.28 ± 0.03 c	32.73 ± 0.13 b	13.57 ± 0.15 b
GP	−0.38 ± 0.03 d	29.25 ± 0.16 d	1.34 ± 0.13 f

Different letters indicate statistically significant differences at *p* ≤ 0.05 according to the results of one-way ANOVA and Tukey’s HSD test. Values are the mean of three technical replicates ± SD.

**Table 10 foods-11-01011-t010:** Main classes of volatile compounds detected in wines tested at racking and their relative percentage distribution. C = Castellina, G = Gaiole, T = Tavarnelle, CP = Castellina plantarum, GP = Gaiole plantarum, and TP = Tavarnelle plantarum.

Constituents	l.r.i.	C	CP	T	TP	G	GP
acetic acid	602	1.2 ± 0.14 a	1.0 ± 0.15 b	0.8 ± 0.13 c	0.7 ± 0.10 c	1.4 ± 0.08 a	1.3 ± 0.12 a
ethyl acetate	603	2.0 ± 0.35 c	1.9 ± 0.32 d	1.8 ± 0.41 d	2.0 ± 0.34 c	2.2 ± 0.33 b	2.6 ± 0.38 a
3-methylbutanol	736	9.1 ± 0.71 a	8.3 ± 0.74 a	6.5 ± 0.75 b	6.6 ± 0.80 b	7.6 ± 0.70 b	8.5 ± 0.74 a
2-methylbutanol	737	4.5 ± 0.52 a	4.4 ± 0.51 a	2.9 ± 0.57 b	3.2 ± 0.58 b	4.3 ± 0.59 a	3.9 ± 0.60 a
2,3-butanediol	790	0.2 ± 0.01 b	0.2 ± 0.01 b	0.2 ± 0.02 b	0.1 ± 0.01 c	0.3 ± 0.02 a	0.3 ± 0.02 a
1,3-butanediol	791	0.9 ± 0.02 c	0.9 ± 0.03 c	0.8±0.01 d	0.7 ± 0.02 e	1.2 ± 0.01 b	1.3 ± 0.02 a
ethyl butyrate	803	0.2 ± 0.02 b	0.2 ± 0.02 b	0.2 ± 0.01 b	0.1 ± 0.02 c	0.1 ± 0.01 c	0.3 ± 0.02 a
1-hexanol	871	0.2 ± 0.01 b	0.2 ± 0.02 b	0.2 ± 0.02 b	0.2 ± 0.01 b	0.3 ± 0.01 a	0.3 ± 0.02 a
isopentyl acetate	877	5.1 ± 0.57 a	4.0 ± 0.52 b	3.3 ± 0.51 c	3.0 ± 0.60 c	2.6 ± 0.57 c	3.3 ± 0.67 c
2-methyl-1-butyl acetate	880	1.5 ± 0.05 a	1.0 ± 0.01 b	0.8 ± 0.02 c	0.7 ± 0.06 d	0.7 ± 0.02 d	0.8 ± 0.03 c
ethyl hexanoate	998	5.8 ± 0.60 a	5.3 ± 0.61 a	4.6 ± 0.65 b	4.8 ± 0.61 b	7.0 ± 0.67 a	6.5 ± 0.64 a
ethyl heptanoate	1098	0.0 ± 0.01 b	0.0 ± 0.01 b	0.2 ± 0.01 a	0.0 ± 0.01 b	0.0 ± 0.01 b	0.0 ± 0.01 b
phenylethyl alcohol	1111	9.0 ± 0.75 a	8.5 ± 0.77 a	7.0 ± 0.73 b	6.8 ± 0.79 b	10.1 ± 0.69 a	9.8 ± 0.75 a
octanoic acid	1179	0.0 ± 0.01 b	0.0 ± 0.01 b	0.1 ± 0.01 a	0.0 ± 0.01 b	0.1±0.01 a	0.0 ± 0.01 b
ethyl octanoate	1197	31.8 ± 0.9 a	31.5 ± 0.86 a	32.7 ± 0.93 a	31.6 ± 0.94 a	33.3 ± 0.86 a	30.9 ± 0.93 a
isopentyl hexanoate	1251	0.0 ± 0.0 b	0.0 ± 0.01 b	0.1 ± 0.01 a	0.0 ± 0.01 b	0.0 ± 0.01 b	0.0 ± 0.01 b
2-phenylethyl acetate	1256	0.6 ± 0.04 a	0.4 ± 0.02 b	0.3 ± 0.01 c	0.3 ± 0.06 c	0.2 ± 0.07 d	0.3 ± 0.03 c
ethyl nonanoate	1296	0.3 ± 0.02 e	0.6 ± 0.04 d	0.8 ± 0.07 c	2.3 ± 0.09 a	0.6 ± 0.04 d	1.2 ± 0.09 b
ethyl 9-decenoate	1389	0.0 ± 0.01 a	0.0 ± 0.01 a	0.1 ± 0.01 a	0.0 ± 0.01 a	0.0 ± 0.01 a	0.0 ± 0.01 a
ethyl decanoate	1397	22.6 ± 0.90 c	25.4 ± 0.93 b	29.4 ± 0.87 a	28.8 ± 0.88 a	23.0 ± 0.85 c	21.1 ± 0.92 c
3-methylbutyl octanoate	1449	0.1 ± 0.02 b	0.2 ± 0.03 a	0.0 ± 0.05 c	0.2 ± 0.01 a	0.1 ± 0.06 b	0.1 ± 0.07 b
2-methylbutyl octanoate	1450	0.0 ± 0.01 c	0.0 ± 0.02 c	0.1 ± 0.02 b	0.2 ± 0.01 a	0.0 ± 0.01 c	0.0 ± 0.01 c
ethyl undecanoate	1496	0.1 ± 0.04 d	0.2 ± 0.07 c	0.2 ± 0.09 c	0.3 ± 0.07 b	0.2 ± 0.06 c	0.4 ± 0.09 a
ethyl dodecanoate	1596	4.7 ± 0.64 d	5.6 ± 0.60 c	6.7 ± 0.66 b	7.3 ± 0.58 a	4.6 ± 0.57 d	7.0 ± 0.62 a
esters		74.8 ± 0.64 c	76.3 ± 0.65 b	81.3 ± 0.67 a	81.6 ± 0.60 a	74.6 ± 0.62 c	74.5 ± 0.66 c
alcohols/phenols		23.9 ± 0.23 a	22.5 ± 0.34 a	17.6 ± 0.44 b	17.6 ± 0.45 b	23.8 ± 0.34 a	24.1 ± 0.54 a
acids		1.2 ± 0.64 a	1.0 ± 0.63 a	0.9 ± 0.67 a	0.7 ± 0.66 a	1.5 ± 0.78 a	1.3 ± 0.76 a
Total identified		99.9	99.8	99.8	99.9	99.9	99.9

Different letters indicate statistically significant differences at *p* ≤ 0.05 according to the results of one-way ANOVA and Tukey’s HSD test. Values are the mean of three technical replicates ± SD.

**Table 11 foods-11-01011-t011:** Main classes of volatile compounds detected in the wines tested at the end of aging and relative percentage distribution.

Constituents	l.r.i.	C	CP	T	TP	G	GP
methyl acetate	528	1.0 ± 0.02 a	0.7 ± 0.03 c	0.0 ± 0.01 e	0.9 ± 0.04 b	0.2 ± 0.02 d	0.9 ± 0.03 b
acetic acid	602	4.2 ± 0.52 a	2.9 ± 0.32 b	3.8 ± 0.48 a	2.2 ± 0.52 b	4.3 ± 0.45 a	2.8 ± 0.42 b
ethyl acetate	603	5.6 ± 0.51 a	5.8 ± 0.48 a	3.2 ± 0.44 c	4.7 ± 0.40 b	3.8 ± 0.50 c	6.2 ± 0.53 a
3-methylbutanol	736	7.5 ± 0.65 b	8.9 ± 0.51 a	8.0 ± 0.52 a	7.7 ± 0.59 b	6.9 ± 0.66 b	8.5 ± 0.63 a
2-methylbutanol	737	4.0 ± 0.52 a	3.6 ± 0.44 a	3.9 ± 0.42 a	4.1 ± 0.62 a	3.4 ± 0.53 a	2.6 ± 0.57 b
1-pentanol	766	0.2 ± 0.02 b	0.2 ± 0.03 b	0.0 ± 0.01 c	0.0 ± 0.01 c	0.2 ± 0.01 b	0.3 ± 0.02 a
2,3-butanediol	790	2.1 ± 0.41 a	1.6 ± 0.35 c	2.1 ± 0.32 b	1.8 ± 0.43 b	1.3 ± 0.45 c	2.7 ± 0.39 a
1,3-butanediol	791	0.5 ± 0.03 b	0.4 ± 0.05 c	0.5 ± 0.04 b	0.5 ± 0.05 b	0.3 ± 0.06 d	0.8 ± 0.07 a
hexanal	802	0.0 ± 0.01 b	0.0 ± 0.01 b	0.3 ± 0.01 a	0.0 ± 0.01 b	0.0 ± 0.01 b	0.0 ± 0.01 b
ethyl butyrate	803	0.1 ± 0.01 a	0.2 ± 0.03 c	0.8 ± 0.01 b	0.2 ± 0.01 c	0.2 ± 0.02 c	0.2 ± 0.01 c
ethyl lactate	813	0.1 ± 0.01 d	0.6 ± 0.01 b	0.3 ± 0.02 c	0.9 ± 0.03 a	0.2 ± 0.04 c	0.7 ± 0.01 b
ethyl 2-methylbutyrate	850	0.2 ± 0.01 b	0.2 ± 0.01 b	0.2 ± 0.02 b	0.3 ± 0.01 a	0.3 ± 0.01 a	0.3 ± 0.02 a
ethyl isovalerate	852	0.3 ± 0.02 c	0.3 ± 0.03 c	0.4 ± 0.02 b	0.5 ± 0.01 a	0.5 ± 0.03 a	0.2 ± 0.01 d
(E)-2-hexenal	856	0.5 ± 0.01 c	0.8 ± 0.05 a	0.3 ± 0.01 d	0.0 ± 0.01 e	0.3 ± 0.03 d	0.7 ± 0.03 b
1-hexanol	871	0.2 ± 0.01 b	0.2 ± 0.02 b	0.2 ± 0.01 b	0.2 ± 0.02 b	0.3 ± 0.02 a	0.2 ± 0.01 b
isopentyl acetate	877	1.8 ± 0.20 a	2.0 ± 0.18 a	1.8 ± 0.15 a	1.9 ± 0.20 a	1.6 ± 0.26 a	1.7 ± 0.18 a
2-methyl-1-butyl acetate	880	0.5 ± 0.02 c	0.5 ± 0.02 c	0.5 ± 0.03 c	0.3 ± 0.02 d	0.6 ± 0.03 b	1.5 ± 0.03 a
(Z)-2-heptenal	962	0.2 ± 0.01 b	0.0 ± 0.01 c	0.3 ± 0.02 a	0.0 ± 0.01 c	0.0 ± 0.01 c	0.0 ± 0.01 c
6-methyl-5-hepten-2-one	987	0.2 ± 0.01 a	0.0 ± 0.01 b	0.0 ± 0.01 b	0.0 ± 0.01 b	0.2 ± 0.01 a	0.0 ± 0.01 b
3-octanone	988	0.3 ± 0.02 c	0.0 ± 0.01 e	1.1 ± 0.01 a	0.0 ± 0.01 e	0.6 ± 0.02 b	0.2 ± 0.01 d
2-octanone	991	0.9 ± 0.01 a	0.0 ± 0.01 c	0.0 ± 0.02 c	0.0 ± 0.01 c	0.4 ± 0.01 b	0.0 ± 0.01 c
ethyl hexanoate	998	4.3 ± 0.31 b	5.4 ± 0.43 a	4.9 ± 0.45 b	4.7 ± 0.32 b	4.8 ± 0.37 b	4.5 ± 0.35 b
heptanoic acid	1081	0.0 ± 0.01 d	0.1 ± 0.01 c	0.0 ± 0.02 d	0.2 ± 0.01 b	0.4 ± 0.01 a	0.1 ± 0.01 c
2-nonanone	1093	0.0 ± 0.01 c	0.0 ± 0.01 c	0.0 ± 0.01 c	0.7 ± 0.02 b	0.9 ± 0.02 a	0.0 ± 0.02 c
nonanal	1102	0.2 ± 0.02 b	0.7 ± 0.02 a	0.1 ± 0.01 c	0.0 ± 0.02 d	0.2 ± 0.03 b	0.0 ± 0.01 d
phenylethyl alcohol	1111	12.4 ± 0.31 a	10.4 ± 0.35 b	10.6 ± 0.37 b	10.6 ± 0.33 b	12.8 ± 0.38 a	10.9 ± 0.31 b
octanoic acid	1179	0.0 ± 0.01 c	0.0 ± 0.01 c	0.1 ± 0.01 b	0.4 ± 0.01 a	0.0 ± 0.01 c	0.0 ± 0.01 c
diethyl succinate	1180	3.4 ± 0.21 c	3.4 ± 0.22 c	5.1 ± 0.15 a	3.5 ± 0.18 c	2.7 ± 0.17 d	4.8±0.16 b
ethyl octanoate	1197	26.4 ± 0.56 a	28.7 ± 0.52 a	24.5 ± 0.45 b	27.5 ± 0.40 a	26.7 ± 0.42 a	28.1±0.50 a
2-phenylethyl acetate	1256	0.4 ± 0.02 a	0.3 ± 0.02 b	0.3 ± 0.01 b	0.3 ± 0.02 b	0.3 ± 0.03 b	0.3 ± 0.02 b
(E)-2-decenal	1260	0.1 ± 0.02 b	0.0 ± 0.02 c	1.3 ± 0.04 a	0.0 ± 0.02 c	1.0 ± 0.02 b	0.0 ± 0.02 c
1-decanol	1272	0.0 ± 0.02 e	0.0 ± 0.02 e	1.7 ± 0.04 a	0.8 ± 0.02 b	0.3 ± 0.02 c	1.0 ± 0.02 d
2-undecanone	1294	0.0 ± 0.01 b	0.0 ± 0.01 b	0.0 ± 0.01 b	0.0 ± 0.01 b	0.4 ± 0.01 a	0.0 ± 0.01 b
ethyl nonanoate	1296	0.0 ± 0.01 c	0.2 ± 0.03 b	0.2 ± 0.03 b	0.3 ± 0.03 a	0.0 ± 0.01 c	0.2 ± 0.01 b
1-nonyl acetate	1310	1.6 ± 0.10 b	1.2 ± 0.14 c	0.0 ± 0.16 d	2.6 ± 0.10 a	1.2 ± 0.09 c	0.0 ± 0.01 d
(E)-2-undecenal	1364	0.7 ± 0.06 b	0.0 ± 0.01 c	3.0 ± 0.03 a	0.0 ± 0.01 c	0.5 ± 0.03 b	0.0 ± 0.04 c
ethyl decanoate	1397	17.6 ± 0.50 b	19.1 ± 0.51 a	18.0 ± 0.48 b	18.4 ± 0.45 b	18.2 ± 0.40 b	15.5 ± 0.47 c
1-decyl acetate	1409	0.1 ± 0.02 b	0.0 ± 0.02 d	0.0 ± 0.02 d	0.0 ± 0.02 d	1.5 ± 0.02 a	0.3 ± 0.02 c
3-methylbutyl octanoate	1449	0.0 ± 0.02 c	0.1 ± 0.03 b	0.0 ± 0.01 c	0.0 ± 0.01 c	0.0 ± 0.01 c	1.2 ± 0.02 a
ethyl dodecanoate	1596	2.2 ± 0.23 b	1.4 ± 0.33 c	2.4 ± 0.21 b	3.6 ± 0.26 a	2.3 ± 0.27 b	2.5 ± 0.28 b
esters		65.6 ± 0.54 b	70.1 ± 0.54 a	62.6 ± 0.55 c	70.6 ± 0.48 a	65.1 ± 0.52 b	69.1 ± 0.57 a
alcohols/phenols		26.9 ± 0.43 a	25.3 ± 0.46 b	27.0 ± 0.47 a	25.7 ± 0.42 b	25.5 ± 0.49 b	27.0 ± 0.50 a
acids		4.2 ± 0.32 a	3.0 ± 0.36 b	3.9 ± 0.32 a	2.8 ± 0.39 b	4.7 ± 0.40 a	2.9 ± 0.31 b
aldehydes/ketones		3.1 ± 0.22 c	1.5 ± 0.25 d	6.4 ± 0.32 a	0.7 ± 0.23 e	4.5 ± 0.45 b	0.9 ± 0.35 e
Total identified		99.8	99.9	99.9	99.8	99.8	99.9

Different letters indicate statistically significant differences at *p* ≤ 0.05 according to the results of one-way ANOVA and Tukey’s HSD test. Values are the mean of three technical replicates ± SD.

## Data Availability

Data is contained within the article.

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
