# Peer review of "Technological Improvements on FML in the Chianti Classico Wine Production: Co-Inoculation or Sequential Inoculation?"

_foods, 2022, doi:10.3390/foods11071011_

Round 1

Reviewer 1 Report

Dear authors and editor,

The subject of the present work is really interesting and so far well written BUT is not finished yet as there is no Discussion and no Conclusion. Please add the relative text in order to proceed to more detailed review.

Author Response

REFEREE 1

Dear authors and editor,

The subject of the present work is really interesting and so far well written BUT is not finished yet as there is no Discussion and no Conclusion. Please add the relative text in order to proceed to more detailed review.

Thanks to the referee for taking the time to review our work. From the guidelines to the authors of foods it is clearly indicated that the results can be presented as a single chapter (results and discussion) and the conclusions are optional. However, we agree that in order to better present the conclusions of such a large and structured work, it is necessary to add a small concluding paragraph. Therefore, as suggested by the reviewer we have added the conclusions. This change has been highlighted in yellow in the text.

Reviewer 2 Report

  1. at pag. 4, line 152 please provide reference for the AVN parameter or define this parameter;
  2. at pag. 4, line 153 please provide OIV codes for: alcohol content (%V/V), AVN (net volatile acidity (g/L acetic acid)), dry extract (g/L) and ash (g/L) to the references chapter;
  3. at pag. 9, lines 252-253 please provide reference for the Total lactic acid parameter or define this parameter compared to the one in table 5 (Lactic acid racking);
  4. at pag. 9, lines 252-253 please provide reference for the Δ hexoses parameter or define this parameter;
  5. at pag. 9, lines 252-253 please explain why malolactic fermentation for CP TP and PG is longer then the alcoholic fermentation for the homolactic variants compared to the duration of fermentation (days) in table 3;

6. at pag. 14-16, lines 378-382 provide quantities and SD for all compounds in tables 10 and 11 in order to support the statistical observations.

Author Response

REFEREE 2

We thank the reviewer for taking the time to review our work. Your recommendations have been followed and the corrections have been highlighted in green in the text. Please find all the modifications listed below.

1.at pag. 4, line 152 please provide reference for the AVN parameter or define this parameter;

We thank the reviewer. AVN refers to the net volatile acidity expressed in g/L of acetic acid and the method for the determination is the OIV method marked in the bibliographical references (line 155)

2. at pag. 4, line 153 please provide OIV codes for: alcohol content (%V/V), AVN (net volatile acidity (g/L acetic acid)), dry extract (g/L) and ash (g/L) to the references chapter;

Done (lines 155-157)

3. at pag. 9, lines 252-253 please provide reference for the Total lactic acid parameter or define this parameter compared to the one in table 5 (Lactic acid racking);

We replaced total lactic acid with lactic acid at the end of the malolactic fermentation and then we corrected lactic acid racking with lactic acid at racking in order to improve the clarity (lines 244, Tab 5)

4. at pag. 9, lines 252-253 please provide reference for the Δ hexoses parameter or define this parameter;

It refers to the difference between the initial hexoses and the hexoses at the end of the malolactic fermentation. We added the explanation (lines 254, 255, tab 6).

5. at pag. 9, lines 252-253 please explain why malolactic fermentation for CP TP and PG is longer then the alcoholic fermentation for the homolactic variants compared to the duration of fermentation (days) in table 3;

The referee is right. We added the explanation, (lines 261-265)

6. at pag. 14-16, lines 378-382 provide quantities and SD for all compounds in tables 10 and 11 in order to support the statistical observations.

Done. We have added the SD and ANOVA in tables 10 and 11

Reviewer 3 Report

The authors present an article with an interesting result. The article is well thought out and executed, however it needs some minor changes to improve and complete.

Lines 57 and 60 – In the introduction part, the Latin name is enough, while the information about commercial name od producers should be specified in material and methods part.

Table 1. I suggest to shorten the title of the table (for example - Table 1. Chemical parameters of the grape at harvest), to move text about statistic test in the legend (under the table) and to explain in legend the meaning of abbreviations (C, T, G). Similar should be used for all tables.

Line 168 – Please, check is sensory analysis described in detail in literature cited under 26 or 27?

Lines 184 -185, 202-203 … Instead ethanol trend as a function of the inoculation method and the progress of the sugar content better to use Kinetics of sugars consumption and ethanol production. Trend is not the best choice of word. Better to say kinetics of ’sugar consumption’, ‘ethanol production’ or ‘changes in pH value, acidity and ash during fermentation’ …. Please replace through whole manuscript.

Line 230 - L. plantarum italic

Line 259 -My suggestion is to replace kinetic difference with conversion rate

Figure 3. Please check and correct the figure title. Aging instead of ripening.

Tables 10 and 11 are incomplete. Authors should add the standard deviation value and the ANOVA significance level value in one column and/or Tukey test in all columns (where ANOVA is satisfied). My suggestion is also to expand these tables with the odor threshold levels (OTLs) for the compounds that reached a concentration above their odor threshold (OTL > 1). It should be interesting to include this in discussion of results.

Author Response

REFEREE 3

The authors present an article with an interesting result. The article is well thought out and executed, however it needs some minor changes to improve and complete.

We thank the reviewer for taking the time to review our work. The changes suggested have been highlighted in purple in the text and are listed below.

Lines 57 and 60 – In the introduction part, the Latin name is enough, while the information about commercial name od producers should be specified in material and methods part.

The referee is right. We have eliminated the unhelpful explanations in this part (lines 61, 64)

Table 1. I suggest to shorten the title of the table (for example - Table 1. Chemical parameters of the grape at harvest), to move text about statistic test in the legend (under the table) and to explain in legend the meaning of abbreviations (C, T, G). Similar should be used for all tables.

We thank the referee. We have modified all the captions of the tables according to the Referee's suggestion

Line 168 – Please, check is sensory analysis described in detail in literature cited under 26 or 27?

We checked. the bibliographic reference is right, so 27

Lines 184 -185, 202-203 … Instead ethanol trend as a function of the inoculation method and the progress of the sugar content better to use Kinetics of sugars consumption and ethanol production. Trend is not the best choice of word. Better to say kinetics of ’sugar consumption’, ‘ethanol production’ or ‘changes in pH value, acidity and ash during fermentation’ …. Please replace through whole manuscript.

Thank you for your suggestion. We have replaced the sentence according to  the Referee's remark (lines 188, 189, 204, 205, 207)

Line 230 - L. plantarum italic

Done (line 235)

Line 259 -My suggestion is to replace kinetic difference with conversion rate

Done (line 272)

Figure 3. Please check and correct the figure title. Aging instead of ripening.

Done (line 269)

Tables 10 and 11 are incomplete. Authors should add the standard deviation value and the ANOVA significance level value in one column and/or Tukey test in all columns (where ANOVA is satisfied). My suggestion is also to expand these tables with the odor threshold levels (OTLs) for the compounds that reached a concentration above their odor threshold (OTL > 1). It should be interesting to include this in discussion of results.

We have added the standard deviation and the ANOVA in both tables. With regard to the threshold values, these cannot be added because the GC/MS values are expressed in relative percentages and it is not possible to effectively quantify the various descriptors. Furthermore, we would like to highlight that the aroma of the wine is very complex with a lot of different volatile compounds interacting with each other. In this context synergistic together with masking effects can be generally evidenced and specific threshold values are not useful to represent the whole aromatic profile. For this reason, we used sensory analysis in combination with GC-MS to better describe, through the senses, the final perception of the aromas identified.